# Prevalence and associated factors of depression among adult refugees in East Africa: A protocol for systematic review and meta-analysis

**Abdu Hailu Shibeshi** [1] *, **Abdulkerim Hassen Moloro** [2], **Bizunesh Fantahun Kase** [3], **Betel Zelalem Wubshet** [2], **Abubeker Alebachew Seid** [2]

1 Department of Statistics, College of Natural and Computational Sciences, Samara University, Samara, Ethiopia, 2 Department of Nursing, College of Medicine and Health Sciences, Samara University, Samara, Ethiopia, 3 Department of Public Health, College of Medicine and Health Sciences, Samara University, Samara, Ethiopia

* abduhailu01@gmail.com, abduhailu2022@su.edu.et

## Abstract

### Introduction

Depression is a common mental health problem among adult refugees in East Africa. Refugees in this region face numerous challenges, including mental health issues, with depression being one of the most common and debilitating conditions. However, there is a lack of systematic reviews and meta-analyses that have been conducted to determine the pooled prevalence and associated factors of depression among adult refugees in East Africa. This study aims to assess the prevalence of depression and its associated factors among adult refugees in East Africa through an analysis of pooled data.

### Materials and methods

A comprehensive search of MEDLINE, Cumulative Index to Nursing and Allied Health Literature (CINAHL), PsycINFO, and African Journals Online (AJOL) will be conducted to identify studies that have reported the prevalence and associated factors of depression among adult refugees in East Africa. A checklist from the Joanna Briggs Institute (JBI) will be used to assess the methodological quality. The search will be limited to studies published in English. Data extraction, critical appraisal, and screening of all the retrieved articles will be conducted by two independent researchers. Statistical analysis will be performed using STATA-17 software packages. The meta-analysis will be conducted using a random-effects model. The heterogeneity of the studies will be assessed using the $I^2$ statistic. The publication bias will be assessed using the funnel plot and Egger's test. For determinants of depression, an effect size with a 95% confidence interval will be reported.

### Ethics and dissemination

Ethical approval and informed consent are not required as this is a systematic review of existing publications. The final results will be published in a peer-reviewed journal and presented at national and international conferences.

**Data Availability Statement:** No datasets were generated or analyzed during the current study. All

relevant data from this study will be made available upon study completion.

**Funding:** The author(s) received no specific funding for this work.

**Competing interests:** The authors have declared that no competing interests exist.

**Abbreviations:** AJOL, African Journals Online; CINAHL, Cumulative Index to Nursing and Allied Health Literature; JBI, Joanna Briggs Institute; LGBTQ+, Lesbian, Gay, Bisexual, Transgender and Intersex; LMIC, Low- and Middle-Income Countries; MeSH, Medical Subject Headings; PRISMA-P, Preferred Reporting Items for Systematic Review and Meta-Analysis Protocols; PTSD, Post-Traumatic Stress Disorder.

**PROSPERO registration number:** CRD42024496728

## Introduction

Depression is a common mental disorder characterized by persistent sadness, loss of interest, and decreased energy, significantly impacting individuals daily lives [1, 2]. Globally, over 260 million individuals are affected, with refugees experiencing depression at rates 2–4 times higher than the general population [3]. Refugees in East Africa face a substantial mental health burden due to prolonged displacement, exposure to chronic stressors, and inadequate living conditions in camps, such as food insecurity and limited resources [4–6]. Addressing this disparity is crucial, especially in East Africa, which hosts millions of refugees fleeing conflicts in South Sudan, Somalia, and Ethiopia [2].

The mental health burden among East African refugees is profound, with depression significantly affecting their integration, well-being, and productivity. Studies estimate depression prevalence in this population to range from 20% to 55% [7, 8], with differences attributed to study designs, methodologies, and the characteristics of refugee populations. Key risk factors include gender-based violence [6], lower educational attainment, prolonged displacement [9], and poverty [10]. Women are disproportionately affected [11], as they often face gender-specific challenges such as unequal access to resources and exposure to violence [12, 13]. Additionally, urban refugee camps, which lack robust support networks, are associated with higher depression rates than rural camps [14, 15].

Stigma and limited access to mental health services further exacerbate the challenges faced by refugees [16]. Cultural beliefs and societal discrimination discourage help-seeking behaviors, making early diagnosis and treatment difficult [17]. Moreover, many East African refugee camps are under-resourced, with insufficient infrastructure to address the rising mental health needs of displaced populations. Compounding these issues is the scarcity of reliable data, particularly from smaller or conflict-affected nations like South Sudan and Somalia, which remain underrepresented in the literature [18].

Variations in diagnostic tools and assessment methods across studies complicate efforts to compare and synthesize findings. For example, diagnostic inconsistencies and the underreporting of mental health issues may lead to biased estimates of depression prevalence [19, 20]. In addition, unique stressors faced by specific refugee groups, such as female heads of households and Lesbian, Gay, Bisexual, Transgender and Intersex (LGBTQ+) individuals, are often overlooked in current studies [21–23]. Addressing these gaps is essential for developing comprehensive mental health strategies tailored to diverse refugee populations in East Africa.

This systematic review and meta-analysis aim to provide accurate estimates of depression prevalence and identify key risk factors among adult refugees in East Africa. By synthesizing available evidence, the study seeks to inform the development of culturally relevant interventions and policies that improve access to mental health services, address the specific needs of vulnerable groups, and enhance the overall resilience of refugees. The findings will highlight gaps in current mental health support systems, guide future research, and offer actionable insights for stakeholders, including policymakers and humanitarian organizations, to improve the well-being of displaced populations in the region.

### Review questions

The following review questions provide a framework for this systematic review and meta-analysis:

 i. What is the pooled prevalence of depression among adult refugees in East Africa?

 ii. What are the associated factors of depression and their effect sizes among adult refugees in East Africa?

## Materials and methods

### Protocol registration and reporting of the review findings

This systematic review and meta-analysis will be conducted following the Preferred Reporting Items for Systematic Reviews and Meta-Analyses (PRISMA-P) statement [24]. The protocol was registered in the International Prospective Register of Systematic Reviews PROSPERO registration number: CRD42024496728. The PRISMA flow chart for reporting systematic review and meta-analysis is presented in Fig 1.

### Inclusion and exclusion criteria

The inclusion criteria specify that only studies involving adult refugees aged 18 years or older will be considered. Articles published in English up to July 2024 will be included, ensuring

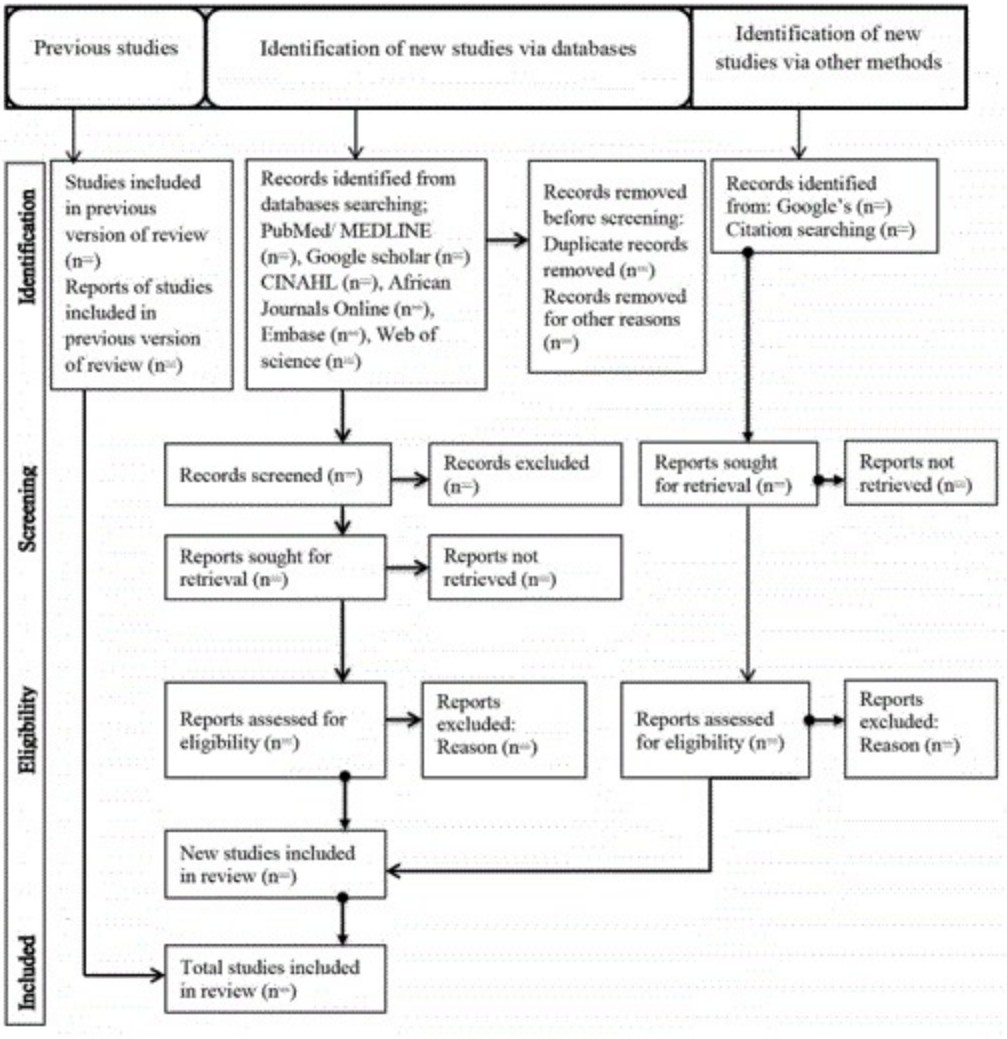

**Fig 1. PRISMA (Preferred Reporting Items for Systematic Reviews and Meta-Analyses) flowchart.**

accessibility and relevance to the review period. The eligible study designs are observational studies, including cross-sectional studies, cohort studies, and population-based surveys, as these provide critical data for assessing the prevalence and associated factors of depression. Additionally, to maintain a regional focus, studies conducted in East African countries such as Burundi, Comoros, Djibouti, Eritrea, Ethiopia, Kenya, Madagascar, Malawi, Mauritius, Mozambique, Rwanda, Seychelles, Somalia, South Sudan, Tanzania, Uganda, Zambia, and Zimbabwe will be included.

On the other hand, the **exclusion criteria** eliminate studies published in languages other than English, as well as conference proceedings, interventional studies, qualitative studies, commentary, editorial letters, case reports, case series, and monthly or annual police reports. The rationale for excluding interventional studies is that they focus on evaluating specific interventions, which is outside the scope of this review, which is primarily concerned with estimating the prevalence of depression and its associated risk factors. Similarly, qualitative studies are excluded because they do not provide the quantitative data required for meta-analysis, which is central to synthesizing pooled estimates in this review.

## Search guide

The "**CoCoPop** search guide" framework will be used to develop a systematic and comprehensive search strategy, ensuring alignment with the study's objectives [25]. This framework focuses on three core components:

- **Condition/Outcome**: Depression

- **Population**: All adult refugee populations

- **Context/Setting**: East Africa

## Measurement outcome variables

The purpose of this review is to assess two outcomes: the first is to determine the pooled prevalence of depression among adult refugees in East Africa, and the second is to identify the factors associated with depression.

## Search strategy and sources of information

A comprehensive search will be conducted in electronic databases, including MEDLINE, Cumulative Index to Nursing & Allied Health Literature (CINAHL), African Journals Online (AJOL), and PsycINFO databases. Additional search strategies will include: contacting authors of relevant studies to identify unpublished or gray literature, manually searching the reference lists of identified studies, and consulting with experts in the field for recommendations.

The search strategy will be adapted for each database using Medical Subject Headings (Mesh), keywords, and free text search terms are all employed in PubMed advanced searching and controlled vocabulary terms related to depression, refugees, East Africa, and related concepts and merged using Boolean operators as search phrases and the additional search strategies are detailed insupporting information (**S1 File**). The reference management software (Endnote™) will then import the electronic database search results and eliminate any duplicates.

## Selection of studies

Two researchers (AHS and AAS) will meticulously assess retrieved articles for inclusion through a three-stage review: title screening for relevance, abstract screening against

predefined criteria, and full-text analysis of shortlisted studies. Using Microsoft Excel™, they will collaboratively make decisions, requiring unanimous approval for inclusion. Disagreements will be resolved via discussion or consultation with a third reseracher (AHM). Finally, a definitive list of articles for data extraction will be compiled, along with clear justifications for excluded studies. This rigorous process ensures the selection of only the most relevant and fitting studies for further analysis.

## Data extraction and management

Two independent researchers (AHS and AAS) will extract relevant data from eligible articles using a standardized format. This format mirrors the JBI data extraction form for systematic reviews and research syntheses [26, 27] and has been meticulously developed to capture all key information. To ensure consistency and accuracy, the reviewers will first pilot test the data extraction process on a subset of articles in Microsoft Excel before proceeding with the full dataset.

To comprehensively analyze each included article, the data extraction tool will include: study characteristics (first author, publication year, country, study setting, study design, sample size), population characteristics (demographics, refugee status, duration of displacement), depression assessment tool, prevalence of depression, and factors associated with depression (adjusted odds ratios or other effect estimates). Disagreements during extraction will be actively resolved for consensus, with a third researcher (AHM) consulted if needed, ensuring the highest accuracy and consistency in capturing all crucial information.

## Quality assessment

Two independent researchers (AHS and AAS) will assess each study using the Joanna Briggs Institute (JBI) checklists, which evaluate nine key parameters: (1) an appropriate sampling frame, (2) an appropriate sampling technique, (3) an adequate sample size, (4) clear description of subjects and setting, (5) adequate data analysis, (6) the use of valid methods for the identified conditions, (7) valid measurement for all participants, (8) the use of appropriate statistical analysis, and (9) an adequate response rate. Each parameter will be rated on a binary scale (Yes = 1, No = 0), with a score of 5 or higher indicating a high-quality study suitable for inclusion [28, 29]. In case of disagreements, a third researcher (AHM) will mediate to reach a consensus.

## Data synthesis and statistical analysis

To unveil the prevalence of depression among adult refugees and its potential links to related factors, a rigorous data analysis pipeline will be implemented. STATA 17, seamlessly integrated with Metan, will be the workhorse for data entry and analysis. The random-effects model [30], coupled with the Freeman-Tuckey transformation, will accurately estimate the pooled prevalence while accounting for potential study variations [31, 32]. Measuring heterogeneity based on statistical findings, outcome presentations, and methodologies will be done using the $I^2$ statistic and a chi-squared test following Cochran's Q statistic with a 5% significance level [33]. $I^2$ values of 25%, 50%, and 75% are considered indicative of low, moderate, and high heterogeneity, respectively [34]. When $I^2 > 50\%$ and the p-value is less than 0.05, the existence of heterogeneity will be declared [34].

Subgroup analyses and meta-regressions will be conducted to explore sources of heterogeneity among the included studies [35]. The proposed variables for these analyses include sex (female vs. male), social support (poor social support vs. good social support), and cumulative traumatic events ($\geq 8$ vs. $< 8$). These factors are essential for understanding the variability in

depression prevalence and associated risk factors among refugee populations. Meta-regression will be used to assess how these variables, along with other key factors such as age, duration of displacement, and access to mental health services, contribute to the heterogeneity observed across studies. Fixed-effect models will be applied when no significant heterogeneity is detected, while random-effects models will be used in cases of expected heterogeneity. When the p-value is less than 0.05, the statistical significance level for effect size is determined. Sensitivity analyses will also be performed to determine the impact of individual studies on pooled estimates, ensuring the robustness and reliability of the findings.

## Publication bias

Publication bias will be assessed using visual inspection of funnel plots and Egger's regression test [36]. It will be considered present if the p-value from Egger's regression test is statistically significant (p < 0.05). Assessing publication bias with a funnel plot requires a minimum of 10 studies [37]. The trim-and-fill method suggested by Duval and Tweedie will be applied if there is proof of publication bias [37, 38].

## Risk of bias assessment

Regarding the quality of the included studies, the researchers will assess them using the JBI checklist for both cross-sectional and cohort studies [39].

# Discussion

This systematic review and meta-analysis aim to address the critical research question of the prevalence of depression among adult refugees in East Africa and identify the associated factors that contribute to this mental health burden. The primary objective of this review is to provide a comprehensive understanding of the factors influencing depression in this population, focusing on key variables such as sex, social support, and cumulative traumatic events. By synthesizing available evidence, the review will offer valuable insights into the mental health challenges faced by refugees in East Africa, guiding future interventions and policy development to improve their well-being.

The justification for conducting this review lies in the significant mental health burden experienced by refugees in the region. Depression is a prevalent condition that affects a substantial proportion of refugees, and understanding its contributing factors is essential for designing targeted interventions. Despite the existing literature, there is a need for a rigorous synthesis of evidence to identify consistent patterns and to better inform mental health strategies.

A summary of the existing literature suggests that depression among refugees in East Africa is influenced by a range of factors, including gender, social support, and exposure to traumatic events. However, the heterogeneity across studies due to varying study designs and geographical locations presents challenges in synthesizing the findings. This review will address this by applying robust statistical techniques, including subgroup analyses and meta-regressions, to explore sources of heterogeneity and provide more reliable estimates.

The methodology for this review is carefully designed to include only observational studies, focusing on those that meet predefined eligibility criteria. Studies will be selected based on their relevance to the research question, with a detailed search strategy employed across databases such as PubMed, CINHAL, and PsycINFO databases. However, certain limitations should be noted. The search strategy will exclude databases such as EMBASE and Web of Science due to access limitations in Ethiopia. Additionally, only articles published in English will be considered, which may result in the exclusion of relevant studies published in other

languages. Furthermore, only observational studies will be included, while randomized clinical trials and quasi-experimental studies will not be part of this review.

Despite these limitations, we are confident that the findings from this review will contribute significantly to the understanding of depression among refugees in East Africa. The results will provide crucial evidence to inform mental health interventions and policy decisions aimed at alleviating the mental health burden in refugee populations. The findings will also be disseminated through peer-reviewed publications and presentations at relevant conferences, ensuring they reach stakeholders involved in refugee health and policy development.

In conclusion, this systematic review and meta-analysis will offer a comprehensive understanding of the prevalence and factors associated with depression in adult refugees in East Africa. By addressing existing gaps in the literature and applying rigorous methodologies, this review will provide valuable insights to guide effective interventions and improve the mental health outcomes of refugees in the region.

## Strengths and limitations of this study

This systematic review will have several strengths. First, the process of data extraction, quality assessment, and eligibility determination for included studies will be conducted independently by two authors to minimize bias and ensure accuracy. Second, the JBI tools will be employed to assess the quality of the eligible studies, ensuring a rigorous evaluation of the included research. A potential limitation of this systematic review is the absence of sufficient meta-analysis studies specifically on the prevalence of depression among adult refugees in East Africa, which may limit the depth of pooled statistical estimates. However, this review will synthesize available observational data to provide meaningful insights despite the lack of meta-analysis studies in this area.

## Supporting information

**S1 File. Searching strategies.**
(DOCX)

**S2 File. PRISMA-P 2015 checklist.**
(DOCX)

## Author Contributions

**Conceptualization:** Abdu Hailu Shibeshi, Abdulkerim Hassen Moloro, Bizunesh Fantahun Kase, Betel Zelalem Wubshet, Abubeker Alebachew Seid.

**Data curation:** Abdu Hailu Shibeshi, Abdulkerim Hassen Moloro, Abubeker Alebachew Seid.

**Formal analysis:** Abdu Hailu Shibeshi, Abdulkerim Hassen Moloro, Bizunesh Fantahun Kase, Abubeker Alebachew Seid.

**Funding acquisition:** Abdu Hailu Shibeshi, Abdulkerim Hassen Moloro, Bizunesh Fantahun Kase, Abubeker Alebachew Seid.

**Investigation:** Abdu Hailu Shibeshi, Abdulkerim Hassen Moloro, Bizunesh Fantahun Kase, Abubeker Alebachew Seid.

**Methodology:** Abdu Hailu Shibeshi, Abdulkerim Hassen Moloro, Bizunesh Fantahun Kase, Abubeker Alebachew Seid.

**Project administration:** Abdu Hailu Shibeshi, Abdulkerim Hassen Moloro, Bizunesh Fantahun Kase, Betel Zelalem Wubshet, Abubeker Alebachew Seid.

**Resources:** Abdu Hailu Shibeshi, Abdulkerim Hassen Moloro, Bizunesh Fantahun Kase, Betel Zelalem Wubshet, Abubeker Alebachew Seid.

**Software:** Abdu Hailu Shibeshi, Abdulkerim Hassen Moloro, Abubeker Alebachew Seid.

**Supervision:** Abdu Hailu Shibeshi, Bizunesh Fantahun Kase, Abubeker Alebachew Seid.

**Validation:** Abdu Hailu Shibeshi, Abdulkerim Hassen Moloro, Bizunesh Fantahun Kase, Abubeker Alebachew Seid.

**Visualization:** Abdu Hailu Shibeshi, Abdulkerim Hassen Moloro, Bizunesh Fantahun Kase, Abubeker Alebachew Seid.

**Writing – original draft:** Abdu Hailu Shibeshi.

**Writing – review & editing:** Abdu Hailu Shibeshi, Abubeker Alebachew Seid.

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
