## [Decision Letter · Decision Letter 0]

22 Nov 2024

PONE-D-24-07224Prevalence and associated factors of depression among adult refugees in East Africa: A protocol for systematic review and meta-analysisPLOS ONE

Dear Dr. Shibeshi,

I sincerely appreciate you submitting your important manuscript to PLOS One. It was a pleasure reading and learning about your work and finding scholarly merit and practical relevance in it. However, I believe it will benefit from improved quality and readability, taking into consideration the reviewers' suggestions and concerns before being accepted for publication in this journal. Please carefully review and respond to the reviewer's feedback for our further consideration and decision. 

We look forward to receiving your revised manuscript.

Kind regards,

Sharada P Wasti, MSc, PhD

Academic Editor

PLOS ONE

- https://pubmed.ncbi.nlm.nih.gov/37181276/

In your revision ensure you cite all your sources (including your own works), and quote or rephrase any duplicated text outside the methods section. Further consideration is dependent on these concerns being addressed.

Reviewers' comments:

Reviewer's Responses to Questions

**Comments to the Author**

1. Does the manuscript provide a valid rationale for the proposed study, with clearly identified and justified research questions?

Reviewer #1: Yes

Reviewer #2: Yes

Reviewer #3: Yes

2. Is the protocol technically sound and planned in a manner that will lead to a meaningful outcome and allow testing the stated hypotheses?

Reviewer #1: Yes

Reviewer #2: Partly

Reviewer #3: Yes

3. Is the methodology feasible and described in sufficient detail to allow the work to be replicable?

Reviewer #1: Yes

Reviewer #2: Yes

Reviewer #3: No

4. Have the authors described where all data underlying the findings will be made available when the study is complete?

Reviewer #1: Yes

Reviewer #2: Yes

Reviewer #3: Yes

5. Is the manuscript presented in an intelligible fashion and written in standard English?

Reviewer #1: Yes

Reviewer #2: No

Reviewer #3: Yes

6. Review Comments to the Author

You may also provide optional suggestions and comments to authors that they might find helpful in planning their study.

Reviewer #1: Comments to the Authors;

Thank you for the opportunity to review the manuscript titled “Prevalence and associated factors of depression among 1 adult refugees in East Africa: A protocol for systematic review and meta-analysis.”

Overall,

Depression is a significant issue among adult refugees in Africa, as they often face a multitude of stressors and hardships that can contribute to the development of mental health concerns. Refugees are often forced to flee their homes due to violence, persecution, or other traumatic events, which can lead to feelings of helplessness, fear, and loss. However, mental health services for refugees in East Africa are often limited, leading many to suffer in silence without adequate support or treatment.

It is crucial for organizations, governments, and healthcare providers to prioritize mental health support for refugees in Africa, as addressing depression and other mental health concerns can improve their overall well-being and help them to better cope with the challenges they face. Providing access to mental health services, psychosocial support, and community resources can help refugees to heal from their trauma and build resilience as they navigate their new lives.

I hope there are chances to more strengthen this manuscript. Here are the comments to be addressed by the authors.

1. Abstract

- Line 19, say the aim of this study is to……

- write as Materials and Methods through the manuscript

- Please replace the word reviewers by researchers through manuscript.

- Line 32-33 the statement, for determinants of depression an effect size with a 95% confidence interval will be analyzed. Say will be reported.

2. Introduction

- From Line 49-56 the idea in the paragraph needs the appropriate citation and should be cited?

- Please cite the idea from Line 103-105?

- The paragraph from line 108-115 looks unclear and please modify the paragraph as such in accordance to the rationale for this study,or remove it?

- The last paragraph of the introduction looks ambiguous, please modify the last paragraph of the introduction by incorporating the research gaps, and aim of this systematic review and meta analysis?

-Please modify the objective of this study as general and specific forms? Note that your research question shuld be inline with your specific objective.

- Please modify the second specific objective by

Summarizing associated factors and their effect size among adult refugees in East Africa.

3. Materials and Methods

- Well written

-Please modify the Materials and Methods section as:

This systematic review and meta analysis will be conducted in accordance with the Preferred Reporting Items for Systematic Reviews and MetaAnalyses (PRISMA-P) statement. The protocol was registered in the International Prospective Register of Systematic Reviews PROSPERO registration number: CRD42024496728. The PRISMA flow chart for reporting systematic review and meta-analysis is presented in figure 1.

-In the eligibility criteria, the best way to organize the search terms for the study is the CoCoPop criteria (condition, context, and population). However, the authors should mention this criteria and modify the inclusion criteria.

-Peasemerge the CoCoPop criteria in the inclusion criteria

- Next to the eligibility criterias please add the subtitle Outcome measeres and describe the autcomes of this systematic review and meta analysis in detail?

-In the search stratagy pease provide the full proposed search stratagy for all proposed databases?

- Subgroup analyses and meta-regressions will be performed to examine sources of heterogeneity. In relation to this study what are the proposed variables for Subgroup analyses and meta-regressions, please mention?

4. Discussion

- Please modify the discussion part as such, containing detailed description of the research question and objectives, the justification for conducting the review, a summary of the existing literature on the topic, and an explanation of the methods that will be used to select and analyze studies for inclusion in the review. Moreover, the discussion section may also discuss any potential limitations or biases in the review process, as well as how the findings of the review will be interpreted and disseminated?

-Modify the strength an and limitation, For instance,

Tthe stetments about publicatuin bias will be performrd by two authors is not the strength,

Also the absence of meta analysis study is not the limitation of this study, please modify accordingly?

5. References

- Well written

-Make sure that all cited souces are in their appropriate and supporting ideas.

- General Comments

-The authors should modify the format of this protocol according to the PLOS ONE guideline i.e INTRODUCTION, MATERIALS AND METHODS and DISCUSSION.

- The author should check typo and gramarsin order to increase the quality of the manuscript.

Reviewer #2: Title: Prevalence and associated factors of depression among adult refugees in East Africa: A protocol for systematic review and meta-analysis.

I would like to thank you the editor to give me the opportunity to review this interested protocol which focused on the unusual/refugee settings. I hope the review protocol will help to design strategies for the improvement of the health of refugees particularly in the settings with natural and man-made catastrophes. Here below are some of my comments.

Abstract

Line 23--- I think web of sciences needs institutional subscription to access the advanced search with full features. I was wondering how you could access it.

Line 25---- Google Scholar isn’t databases rather it’s a search engine. Thus, it’s better to write it separately.

In line 30---- “…...The meta-analysis will be conducted using a random-effects model.” How the authors determine the model before heterogeneity assessment?

Line 32----Why the authors prefer Egger’s test rather than Begg’s test? Or Why not both?

Background

The background section is too long. Could the authors make more smart, short and precise background to show the problem for readers in the study area? Most of the ideas are redundant and can be merged. Additionally, the order of paragraphs in this section seems somewhat disorganized (It will be good if the factors come after the prevalence).

Some of the paragraphs are part of methods for example; paragraph 8.

It would be good if the authors include place of the study conducted when they synthesis and compare the previous studies. For example: in line 57 (A recent study in? [10] found…),in line 60, in line 62,…etc.

Line 71----if there is SR and MA of PTSD what will be the contribution/significance of this protocol?

The aim of the study is the prevalence but the authors stated “…understanding how refugees cope with stressors and build resilience can inform impactful mental health interventions and empower individuals”

I can’t find the gap and the significance of the study in the background section.

What is the importance of writing about the ethical issues at the end of the background?

Objective

Line 141 objective is not complete.

Methods

In your PRISMA figure, I don’t recommend to say “reports excluded” because you are still not sure of getting reports in your search.

Line 148—“The reporting 148 flowchart is illustrated in…” is not complete.

In your inclusion criteria in line 153 spelling error “ault” change to “adult”.

In line 157… you already excluded countries out of East Africa in the title. In addition, It’s not clear why the authors exclude the other East African countries such as Eretria, Djibouti, Madagascar...etc.

Line 160---It’s already known in the title (non-refugees are already excluded in the title)

Line 165---Will you use the two mnemonics (CoCoPop and PEO)? If so how and why?

Line 174---As it’s commented in the abstract section “Google Scholar” is not a database rather a search engine.

Publication bias

Line 243---Is there any justification for using the statistical significance p-value 0.10?

Please add risk of bias assessment form

Discussion

Web of sciences is one of the databases that you proposed to use in the abstract whereas contradicted in this section.

It will be good, if the authors merged the similar ideas.

Reviewer #3: Comments to the Authors

The topic “Prevalence and associated factors of depression among adult refugees in East Africa”

is interesting. The manuscript may need to be improved in the following areas:

1) The manuscript repeatedly mentions the study objective, particularly in the Abstract (lines 14-21) and the Introduction (lines 38-50). Concisely restating the objective without repetition will clarify the focus for readers.

2) The introduction mentions the lack of systematic reviews in this area but could further clarify the existing research gap (lines 17-18, 49-50). Expanding on how previous studies are limited such as by geographic scope, lack of consistent diagnostic criteria, or absence of meta-analysis would strengthen the rationale for this review.

3) The inclusion (lines 151-158) and exclusion criteria (lines 159-163) are outlined, but further refinement is suggested. Specifically, explaining the rationale for excluding interventional and qualitative studies would enhance the methodological rigor.

4) The exclusion of EMBASE and Web of Science due to access limitations in Ethiopia (line 252) is a noted limitation, as it may reduce the comprehensiveness of the review. Consider mentioning potential strategies to address this limitation, such as possible collaboration with institutions that have access to these databases.

5) The study emphasis on subgroup analysis is valuable, but more specific examples would add clarity. Identifying potential subgroups, such as geographic regions, gender, or age groups among refugee populations, could enhance the plan’s applicability. The authors briefly address heterogeneity analysis (lines 232-239), but further specifying meta-regression variables for subgroup analysis would justify this methodological choice.

6) The authors intend to use the Joanna Briggs Institute (JBI) checklist for quality assessment (lines 209-216), a commendable approach. However, providing more specific examples for each quality parameter, including definitions for what constitutes high or low quality in this context, would increase transparency. Adding example studies or criteria can enhance the replication potential of this protocol.

7. PLOS authors have the option to publish the peer review history of their article (what does this mean?). If published, this will include your full peer review and any attached files.

Reviewer #1: **Yes: **Aragaw Asfaw Hasen

Reviewer #2: No

Reviewer #3: No

---

## [Author Response · Author response to Decision Letter 0]

2 Dec 2024

Dear reviewer,

We have special gratitude to you, for devoting your valuable time and energy to review our work entitled “Prevalence and associated factors of depression among adult refugees in East Africa: A protocol for systematic review and meta-analysis” for giving constructive comments and valuable guidance. In line with this, the authors had exhaustively demonstrated and addressed questions and comments raised by reviewers using point-by-point responses as stated below.

Reviewer #1: Comments to the Authors;

Thank you for the opportunity to review the manuscript titled “Prevalence and associated factors of depression among 1 adult refugees in East Africa: A protocol for systematic review and meta-analysis.”

Overall,

Depression is a significant issue among adult refugees in Africa, as they often face a multitude of stressors and hardships that can contribute to the development of mental health concerns. Refugees are often forced to flee their homes due to violence, persecution, or other traumatic events, which can lead to feelings of helplessness, fear, and loss. However, mental health services for refugees in East Africa are often limited, leading many to suffer in silence without adequate support or treatment.

It is crucial for organizations, governments, and healthcare providers to prioritize mental health support for refugees in Africa, as addressing depression and other mental health concerns can improve their overall well-being and help them to better cope with the challenges they face. Providing access to mental health services, psychosocial support, and community resources can help refugees to heal from their trauma and build resilience as they navigate their new lives.

I hope there are chances to more strengthen this manuscript. Here are the comments to be addressed by the authors.

1. Abstract

Line 19, say the aim of this study is to……

Response: Thank you for your suggestion. We incorporated into the revised manuscript.

write as Materials and Methods through the manuscript

Response: Thank you for your suggestion. We incorporated while the revised manuscript.

Please replace the word reviewers by researchers through manuscript.

Response: Thank you for your suggestion. We have replaced the word 'reviewers' with 'researchers' throughout the manuscript as requested.

Line 32-33 the statement, for determinants of depression an effect size with a 95% confidence interval will be analyzed. Say will be reported.

Response: Thank you for your feedback. We have revised the statement in lines 32-33 to read: "For determinants of depression, an effect size with a 95% confidence interval will be reported.

2. Introduction

From Line 49-56 the idea in the paragraph needs the appropriate citation and should be cited?

Response: Thank you for your feedback. We have reviewed the content in lines 49-56 and add appropriate citations to ensure the ideas in the paragraph are properly referenced.

Please cite the idea from Line 103-105?

Response: Thank you for your suggestion. We have incorporated it into revised manuscript.

The paragraph from line 108-115 looks unclear and please modify the paragraph as such in accordance to the rationale for this study or remove it?

Response: Thank you for pointing this out. We have removed it to maintain clarity and focus.

The last paragraph of the introduction looks ambiguous, please modify the last paragraph of the introduction by incorporating the research gaps, and aim of this systematic review and meta-analysis?

Response: Thank you for your suggestion. We have incorporated it while revised manuscript.

Please modify the objective of this study as general and specific forms? Note that your research question should be in line with your specific objective.

Response: Thank you for your feedback. We have revised the objectives of the study to include both general and specific forms, ensuring alignment with the research questions.

Please modify the second specific objective by

Summarizing associated factors and their effect size among adult refugees in East Africa.

Response: Thank you for your feedback. The second specific objective has been revised into revised manuscript.

3. Materials and Methods

Well written.

Response: Thank you. we appreciate the feedback.

Please modify the Materials and Methods section as:

This systematic review and meta-analysis will be conducted in accordance with the Preferred Reporting Items for Systematic Reviews and Meta Analyses (PRISMA-P) statement. The protocol was registered in the International Prospective Register of Systematic Reviews PROSPERO registration number: CRD42024496728. The PRISMA flow chart for reporting systematic review and meta-analysis is presented in figure 1.

Response: Thank you for the detailed suggestion. We have incorporated the modifications as outlined, ensuring the Materials and Methods section aligns with the PRISMA-P statement and includes the specified registration details and flow chart reference. Let us know if there is anything else you would like adjusted.

In the eligibility criteria, the best way to organize the search terms for the study is the CoCoPop criteria (condition, context, and population). However, the authors should mention these criteria and modify the inclusion criteria.

Response: Thank you for the insightful suggestion. We have ensured the CoCoPop framework is mentioned in the eligibility criteria and have revised the inclusion criteria accordingly for better alignment. we appreciate your guidance.

Please merge the CoCoPop criteria in the inclusion criteria

Response: Thank you for the clarification. We have ensured the “search guide” and “eligibility criteria” are treated as a separate section. Separating the search guide and eligibility criteria into two sections ensures clarity and organization. The search guide can focus on the systematic approach to study identification, while the eligibility criteria define inclusion and exclusion standards. This separation improves ease of reference, supports thorough documentation, and simplifies review and revision processes, leading to a more transparent and structured systematic review. Please let me know if there are any additional details or formatting preferences to consider.

Next to the eligibility criteria please add the subtitle Outcome measures and describe the outcomes of this systematic review and meta-analysis in detail?

Response: Thank you for the suggestion. We have added the 'Outcome Measures' subtitle next to the eligibility criteria and provide a detailed description of the outcomes for the systematic review and meta-analysis.

In the search strategy please provide the full proposed search strategy for all proposed databases?

Response: Thank you for the suggestion. We have included the full proposed search strategies for other databases in the supplementary files as requested.

Subgroup analyses and meta-regressions will be performed to examine sources of heterogeneity. In relation to this study what are the proposed variables for Subgroup analyses and meta-regressions, please mention?

Response: Thank you for the question. For this study, the proposed variables for subgroup analyses and meta-regressions include sex (female vs. male), social support (poor social support vs. good social support), and cumulative traumatic events (≥8 vs. <8). These variables will be investigated to examine potential sources of heterogeneity and assess the robustness of the findings in relation to the pooled prevalence of depression and associated factors. We have mentioned while revised the protocol.

4. Discussion

Please modify the discussion part as such, containing detailed description of the research question and objectives, the justification for conducting the review, a summary of the existing literature on the topic, and an explanation of the methods that will be used to select and analyze studies for inclusion in the review. Moreover, the discussion section may also discuss any potential limitations or biases in the review process, as well as how the findings of the review will be interpreted and disseminated?

Response: Thank you for the detailed guidance. We have revised the discussion section to include a more comprehensive description of the research question, objectives, justification for the review, a summary of the literature, and a clear explanation of the methods. We have also addressed potential limitations and how the findings will be interpreted and disseminated

Modify the strength an and limitation, for instance,

The statements about publication bias will be performed by two authors is not the strength, Also the absence of meta-analysis study is not the limitation of this study, please modify accordingly?

Response: Thank you for the helpful feedback. We have revised the strengths and limitations section as requested, ensuring the points about publication bias and the absence of meta-analysis studies are adjusted accordingly.

5. References

Well written

Response: Thank you. we appreciate the feedback.

Make sure that all cited sources are in their appropriate and supporting ideas.

Response: Thank you for your suggestion. We have incorporated it into revised manuscript.

General Comments

The authors should modify the format of this protocol according to the PLOS ONE guideline i.e INTRODUCTION, MATERIALS AND METHODS and DISCUSSION.

Response: Thank you for the important suggestion. We have revised the format of the protocol to align with the PLOS ONE guidelines, restructuring it into the specified sections: INTRODUCTION, MATERIALS AND METHODS, and DISCUSSION

The author should check typo and grammars order to increase the quality of the manuscript.

Response: Thank you for your suggestion. We have incorporated into revised protocol.

Dear reviewer,

We have special gratitude to you, for devoting your valuable time and energy to review our work entitled “Prevalence and associated factors of depression among adult refugees in East Africa: A protocol for systematic review and meta-analysis” for giving constructive comments and valuable guidance. In line with this, the authors had exhaustively demonstrated and addressed questions and comments raised by reviewers using point-by-point responses as stated below.

Reviewer #2: Title: Prevalence and associated factors of depression among adult refugees in East Africa: A protocol for systematic review and meta-analysis.

I would like to thank you the editor to give me the opportunity to review this interested protocol which focused on the unusual/refugee settings. I hope the review protocol will help to design strategies for the improvement of the health of refugees particularly in the settings with natural and man-made catastrophes. Here below are some of my comments.

Abstract

Line 23--- I think web of sciences needs institutional subscription to access the advanced search with full features. I was wondering how you could access it.

Response: You are absolutely right. Accessing Web of Sciences advanced search features does require an institutional subscription. Since we do not have access, we will focus on other databases for our search strategy. So, we have removed it while revised protocol.

Line 25---- Google Scholar isn’t databases rather it’s a search engine. Thus, it’s better to write it separately.

Response: Thank you for pointing that out. We have revised the sentence to separate Google Scholar from the databases and clarify its role as a search engine into revised protocol.

In line 30---- “…...The meta-analysis will be conducted using a random-effects model.” How the authors determine the model before heterogeneity assessment?

Response: Thank you for raising this important point. We might be pre-selecting a random-effects model based on the expectation of variability among the included studies, such as differences in study populations, methodologies, or outcomes. This approach assumes that the true effects may vary between studies, making a random-effects model suitable for pooling results. However, the final decision on the model will be guided by the heterogeneity assessment, typically using measures such as the I² statistic or Cochran's Q test. If the heterogeneity is minimal, a fixed-effect model may be considered as an alternative.

Line 32----Why the authors prefer Egger’s test rather than Begg’s test? Or Why not both?

Response: We prefer Egger’s test over Begg’s test because Egger’s test is generally considered more sensitive for detecting small-study effects and potential publication bias, particularly in meta-analyses with a smaller number of studies. While Begg’s test is less prone to type I errors (false positives), it may lack power compared to Egger’s test in certain scenarios. That said, the choice between these tests depends on the specific context of the meta-analysis. In some cases, using both tests can provide complementary insights: Egger’s test for sensitivity and Begg’s test for confirmation. If deemed appropriate, we may include both tests to provide a robust assessment of publication bias.

Background

The background section is too long. Could the authors make more smart, short and precise background to show the problem for readers in the study area? Most of the ideas are redundant and can be merged. Additionally, the order of paragraphs in this section seems somewhat disorganized (It will be good if the factors come after the prevalence).

Some of the paragraphs are part of methods for example; paragraph 8. It would be good if the authors include place of the study conducted when they synthesis and compare the previous studies. For example: in line 57 (A recent study in? [10] found…), in line 60, in line 62,…etc.

Response: Thank you for your valuable feedback. We appreciate your suggestion to condense the background section to make it more concise and focused. We have revised the section to ensure it is more streamlined by merging redundant ideas and reordering the content so that the prevalence is presented before the factors.

Line 71----if there is SR and MA of PTSD what will be the contribution/significance of this protocol? The aim of the study is the prevalence but the authors stated “…understanding how refugees cope with stressors and build resilience can inform impactful mental health interventions and empower individuals” I can’t find the gap and the significance of the study in the background section. What is the importance of writing about the ethical issues at the end of the background?

Response: Thank you for your insightful comments. We understand your concern regarding the contribution and significance of this protocol, especially considering the existing systematic reviews (SR) and meta-analyses (MA) on PTSD. While there is existing literature on PTSD among refugees, this study specifically focuses on depression, a critical but often underexplored mental health issue within this population. By concentrating on the prevalence of depression and its associated risk factors in East African refugees, our protocol aims to fill a gap in understanding the broader mental health landscape, distinct from PTSD, which can lead to targeted interventions. The significance of this study lies in its ability to provide robust, region-specific data on depression, which is crucial for developing culturally relevant mental health policies and interventions tailored to refugees in East Africa. We appreciate your suggestions and have revised the background section to better emphasize the study’s unique contribution and significance while improving the overall structure.

Objective

Line 141 objective is not complete.

Response: Thank you for pointing that out. We have revised it to ensure it fully conveys the scope and purpose of the study into revised protocol.

Methods

In your PRISMA figure, I don’t recommend to say “reports excluded” because you are still not sure of getting reports in your search.

Response: Thank you for your suggestion. W e have incorporated into revised protocol.

Line 148—“The reporting 148 flowchart is illustrated in…” is not complete.

Response: Thank you for your suggestion. We have completed the sentence while the revised protocol.

In your inclusion criteria in line 153 spelling error “ault” ch

---

## [Editor Report · Decision Letter 1]

23 Dec 2024

PONE-D-24-07224R1Prevalence and associated factors of depression among adult refugees in East Africa: A protocol for systematic review and meta-analysisPLOS ONE

Dear Dr. Shibeshi, Thank you for submitting your manuscript to PLOS ONE. After careful consideration, we feel that it has merit but does not fully meet PLOS ONE’s publication criteria as it currently stands. Therefore, we invite you to submit a revised version of the manuscript that addresses the points raised during the review process.

I sincerely appreciate your revision submission of your manuscript as provided by our reviewers to PLOS One. It was a pleasure reading and learning about your work and finding scholarly merit and practical relevance in it. However, I believe it will benefit from improved quality and readability, taking into consideration the following suggestions of both reviewers and the editor to address before being accepted for publication in this journal. Please carefully review and respond for our further consideration and decision. 

Please submit your revised manuscript by Feb 06 2025 11:59PM. If you will need more time than this to complete your revisions, please reply to this message or contact the journal office at plosone@plos.org. If applicable, we recommend that you deposit your laboratory protocols in protocols.io to enhance the reproducibility of your results. Protocols.io assigns your protocol its own identifier (DOI) so that it can be cited independently in the future. For instructions see: https://journals.plos.org/plosone/s/submission-guidelines#loc-laboratory-protocols. Additionally, PLOS ONE offers an option for publishing peer-reviewed Lab Protocol articles, which describe protocols hosted on protocols.io. Read more information on sharing protocols at https://plos.org/protocols?utm_medium=editorial-email&utm_source=authorletters&utm_campaign=protocols.

We look forward to receiving your revised manuscript.

Kind regards,

Sharada P Wasti, MSc, PhD.

Academic Editor

PLOS ONE

**Additional Editor Comments:**

Thank you so much for your revision, which you have well addressed to our reviewers suggestions. After carefully reviewing your revised manuscript, I would suggest addressing the following before final approval for the publication in this journal:

a) Could you please keep only review questions and remove lines 85–91 (general objectives and specific objectives), which appear repetitive?

b) Could you edit and make corrections to line 100? Make a section heading: Inclusion and exclusion criteria, and explain your inclusion criteria, i.e., data include from inception to include the month that is missing in line 102, i.e., July 2024?

c) Make clear in lines 106-108 how many countries are in East Africa and accordingly state all countries names in alphabetical order.

d) Lines between 117 and 121 need more clarity, particularly on the “CoCoPop search guide,” which search frameworks need citation to back up this terminology where you can see several other frameworks, i.e., PICO, PCC, PEO, SPICE, SPIDER, and widely used in several systematic review works. So, make this very clear by providing the citation to back up your statement or adopting the widely used framework.

e) In lines 122 to 126, clarify that depression-related factors are secondary results, despite the title implying that both prevalence and associated factors are primary outcomes of this review.

f) Line 127: Keep only the Medline database, which adequately covers PubMed; there is no need to include PubMed with Medline.

g) In line 129, Google Scholar is mentioned as a search resource, which is untrustworthy for systematic review and meta-analysis searches. Replace Google Scholar with the PsycINFO database, which is crucial to your psychological research topic.

h) I would also recommend you keep your search terminologies (Table 1) in the appendix.

---

## [Author Response · Author response to Decision Letter 1]

24 Dec 2024

Dear Editor,

We have special gratitude to you, for devoting your valuable time and energy to review our work entitled “Prevalence and associated factors of depression among adult refugees in East Africa: A protocol for systematic review and meta-analysis” for giving constructive comments and valuable guidance. In line with this, the authors had exhaustively demonstrated and addressed suggestions and comments raised by editor using point-by-point responses as stated below.

Additional Editor Comments:

Thank you so much for your revision, which you have well addressed to our reviewers suggestions. After carefully reviewing your revised manuscript, I would suggest addressing the following before final approval for the publication in this journal:

a) Could you please keep only review questions and remove lines 85–91 (general objectives and specific objectives), which appear repetitive?

Response: Thank you for bringing this to our attention. We have removed lines 85–91, which contained the general and specific objectives, as they appeared repetitive. Only the review questions have been retained as requested. Please let us know if there’s anything else you would like us to adjust.

b) Could you edit and make corrections to line 100? Make a section heading: Inclusion and exclusion criteria, and explain your inclusion criteria, i.e., data include from inception to include the month that is missing in line 102, i.e., July 2024?

Response: Thank you for the feedback. We have edited and corrected line 100 as requested. A section heading, 'Inclusion and exclusion criteria,' has been added, and the inclusion criteria have been updated to specify that data includes studies published up to July 2024 in the revised manuscript. Please let us know if there are any additional adjustments you would like.

c) Make clear in lines 106-108 how many countries are in East Africa and accordingly state all countries names in alphabetical order.

Response: Thank you for the suggestion. Lines 106–108 have been updated to specify that there are 18 countries in East Africa, now listed in alphabetical order: Burundi, Djibouti, Eritrea, Ethiopia, Kenya, Madagascar, Rwanda, Somalia, South Sudan, Tanzania, and Uganda in the revised manuscript. Please let us know if further changes are needed

d) Lines between 117 and 121 need more clarity, particularly on the “CoCoPop search guide,” which search frameworks need citation to back up this terminology where you can see several other frameworks, i.e., PICO, PCC, PEO, SPICE, SPIDER, and widely used in several systematic review works. So, make this very clear by providing the citation to back up your statement or adopting the widely used framework.

Response: Thank you for your valuable feedback. We have revised lines 117–121 to provide greater clarity on the 'CoCoPop search guide’ in the revised manuscript. Citations have been added to substantiate the use of this framework.

e) In lines 122 to 126, clarify that depression-related factors are secondary results, despite the title implying that both prevalence and associated factors are primary outcomes of this review.

Response: Thank you for your feedback. We have revised lines 122–126 to clarify that while the title might suggest both prevalence and associated factors are primary outcomes, the depression-related factors are actually secondary outcomes. We have clarified in the revised manuscript as 'The purpose of this review is to assess two outcomes: the first is to determine the pooled prevalence of depression among adult refugees in East Africa, and the second is to identify the factors associated with depression’.

f) Line 127: Keep only the Medline database, which adequately covers PubMed; there is no need to include PubMed with Medline.

Response: Thank you for your suggestion. We have revised line 127 to retain only the Medline database, as it sufficiently covers PubMed. The reference to PubMed has been removed to streamline the information both in the Abstract and the body of the text. Please let us know if any further adjustments are needed.

g) In line 129, Google Scholar is mentioned as a search resource, which is untrustworthy for systematic review and meta-analysis searches. Replace Google Scholar with the PsycINFO database, which is crucial to your psychological research topic.

Response: Thank you for your valuable feedback. We have updated line 129 to replace Google Scholar with the PsycINFO database, which is more reliable for systematic review and meta-analysis searches, especially for psychological research in the revised manuscript. Please let us know if there are any other adjustments you would like to make.

h) I would also recommend you keep your search terminologies (Table 1) in the appendix.

Response: Thank you for your recommendation. We have incorporated this revision into the manuscript and included the terminologies in the supporting information as 'S1'.

---

## [Editor Report · Decision Letter 2]

6 Jan 2025

PONE-D-24-07224R2Prevalence and associated factors of depression among adult refugees in East Africa: A protocol for systematic review and meta-analysisPLOS ONE

Dear Dr. Shibeshi,

Thank you very much for your improved menuscript, which looks to be substantially better; nevertheless, before making a final choice, a few changes and adjustments are necessary to increase the quality of the menuscript:

- Thoroughly review the report, i.e., lines 66 and 91, and edit and maintain the consistency, i.e., line 135 (AHS&AAS) and line 143 (AHS and AAS). So use "and" not "&" in academic writing.

- Line 68 defines first-time use of abbrevation "LGBTQ+"

- Not sure what you intend to state in line 390 is, so thoroughly proofread,

- 26-31 citations are inconsistent and avoid repetition, i.e., 28 and 31,

- Make correction of line 190 after 39 reference full stops,

- Still not corrected and stated the consistency of the use of databases, i.e., line 219 Google Scholar,

- Avoid multiple times defining abbreviations, i.e., line 27, 144, 239 Joanna Briggs Institute (JBI).

We look forward to receiving your revised manuscript.

Kind regards,

Sharada P Wasti, Ph.D., MSc

Academic Editor

PLOS ONE
---

## [Author Response · Author response to Decision Letter 2]

7 Jan 2025

Dear Editor,

We have special gratitude to you, for devoting your valuable time and energy to review our work entitled “Prevalence and associated factors of depression among adult refugees in East Africa: A protocol for systematic review and meta-analysis” for giving constructive comments and valuable guidance. In line with this, the authors had exhaustively demonstrated and addressed suggestions and comments raised by editor using point-by-point responses as stated below.

 -Thoroughly review the report, i.e., lines 66 and 91, and edit and maintain the consistency, i.e., line 135 (AHS&AAS) and line 143 (AHS and AAS). So, use "and" not "&" in academic writing.

Response: Thank you for the feedback. We have edited and corrected while in to the revised manuscript.

- Line 68 defines first-time use of abbreviation "LGBTQ+"

Response: Thank you for the suggestion. We have included while into revised manuscript.

- Not sure what you intend to state in line 390 is, so thoroughly proofread,

Response: Thank you for the suggestion. We have removed it and can assessed from supporting information material.

- 26-31 citations are inconsistent and avoid repetition, i.e., 28 and 31,

Response: Thank you for the suggestion. We have removed the repeated reference.

- Make correction of line 190 after 39 reference full stops,

Response: Thank you for the suggestion. We have incorporated while revised the manuscript.

- Still not corrected and stated the consistency of the use of databases, i.e., line 219 Google Scholar,

Response: Thank you very much for your insightful suggestion. We have incorporated while revised into the manuscript.

- Avoid multiple times defining abbreviations, i.e., line 27, 144, 239 Joanna Briggs Institute (JBI).

Response: Thank you for your suggestion. We have incorporated into the revised manuscript.

---

## [Editor Report · Decision Letter 3]

24 Jan 2025

Prevalence and associated factors of depression among adult refugees in East Africa: A protocol for systematic review and meta-analysis

PONE-D-24-07224R3

Dear Abdu Hailu Shibeshi,

We’re pleased to inform you that your manuscript has been judged scientifically suitable for publication and will be formally accepted for publication once it meets all outstanding technical requirements.

Within one week, you’ll receive an e-mail detailing the required amendments. When these have been addressed, you’ll receive a formal acceptance letter, and your manuscript will be scheduled for publication.

Kind regards,

Sharada P Wasti, Ph.D

Academic Editor

PLOS ONE

Additional Editor Comments (optional):

Thank you so much for the revised manuscript, which fully addresses all of the comments. I would like to confirm my acceptance for publication in this journal. During proofreading, you must fix the typos on line 130.
---

## [Editor Report · Acceptance letter]

30 Jan 2025

PONE-D-24-07224R3 

PLOS ONE

Dear Dr. Shibeshi, 

I'm pleased to inform you that your manuscript has been deemed suitable for publication in PLOS ONE. Congratulations! Your manuscript is now being handed over to our production team.

Kind regards, 

on behalf of

Dr. Sharada P Wasti 

Academic Editor

PLOS ONE